# Correlation between *Peptacetobacter hiranonis*, the *baiCD* Gene, and Secondary Bile Acids in Dogs

**DOI:** 10.3390/ani14020216

**Published:** 2024-01-09

**Authors:** Bruna Correa Lopes, Chih-Chun Chen, Chi-Hsuan Sung, Patricia Eri Ishii, Luis Fernando da Costa Medina, Frederic P. Gaschen, Jan S. Suchodolski, Rachel Pilla

**Affiliations:** 1Gastrointestinal Laboratory, Department of Small Animal Clinical Sciences, Texas A&M University, College Station, TX 77840, USA; chihchunchen@tamu.edu (C.-C.C.); csung@cvm.tamu.edu (C.-H.S.); pishii@cvm.tamu.edu (P.E.I.); lfcmedina@tamu.edu (L.F.d.C.M.); jsuchodolski@cvm.tamu.edu (J.S.S.); rpilla@cvm.tamu.edu (R.P.); 2Department of Veterinary Clinical Sciences, School of Veterinary Medicine, Louisiana State University, Baton Rouge, LA 70803, USA; fgaschen@lsu.edu

**Keywords:** fecal unconjugated bile acids, canine, *Clostridium hiranonis*, primary bile acids, conversion of bile acids, bile acid metabolism, *bai* operon

## Abstract

**Simple Summary:**

Dysmetabolism of bile acids has been linked to chronic enteropathy in dogs. *Peptacetobacter* (*Clostridium*) *hiranonis* has been described as the major species responsible for converting primary into secondary bile acids in dogs. Moreover, decreased *P. hiranonis* abundance has been linked to chronic enteropathy and antibiotic-induced dysbiosis in dogs and cats. Therefore, this study aimed to investigate further the correlation between *P. hiranonis*, the bacterial gene (*baiCD*) involved in bile acid conversion, and the conversion process per se. Our findings indicate a strong and significant correlation between *P. hiranonis, baiCD,* and the relative concentration of secondary bile acid in dogs.

**Abstract:**

Bile acid metabolism is a key pathway modulated by intestinal microbiota. *Peptacetobacter* (*Clostridium*) *hiranonis* has been described as the main species responsible for the conversion of primary into secondary fecal unconjugated bile acids (fUBA) in dogs. This multi-step biochemical pathway is encoded by the bile acid-inducible (*bai*) operon. We aimed to assess the correlation between *P. hiranonis* abundance, the abundance of one specific gene of the *bai* operon (*baiCD*), and secondary fUBA concentrations. In this retrospective study, 133 fecal samples were analyzed from 24 dogs. The abundances of *P. hiranonis* and *baiCD* were determined using qPCR. The concentration of fUBA was measured by gas chromatography–mass spectrometry. The *baiCD* abundance exhibited a strong positive correlation with secondary fUBA (ρ = 0.7377, 95% CI (0.6461, 0.8084), *p* < 0.0001). Similarly, there was a strong correlation between *P. hiranonis* and secondary fUBA (ρ = 0.6658, 95% CI (0.5555, 0.7532), *p <* 0.0001). Animals displaying conversion of fUBA and lacking *P. hiranonis* were not observed. These results suggest *P. hiranonis* is the main converter of primary to secondary bile acids in dogs.

## 1. Introduction

Bile acid (BA) metabolism is an important pathway involved in regulating metabolic homeostasis and is modulated by intestinal microbiota. These regulations include lipid and glucose metabolism, energy production, and inflammatory signaling [1,2]. In the context of BA metabolism, cholic acid (CA) and chenodeoxycholic acid (CDCA) are generated in the liver through the catabolism of cholesterol. Subsequently, they are conjugated to the amino acids glycine and taurine by the enzyme amino acid N-acyltransferase [3,4]. In dogs, most bile acids are conjugated to taurine [5,6]. Once conjugated, BAs are actively secreted from the liver, passing through the canicular membrane into the gall bladder and ultimately into the intestinal lumen [7]. Alongside dietary lipids and lipid-soluble nutrients, BAs form micellar structures that facilitate absorption by the enterocytes. This process is essential for the proper digestion and absorption of nutrients, reinforcing the maintenance of overall gastrointestinal health. Approximately 95% of BAs undergo reabsorption via the portal vein in a process referred to as enterohepatic circulation [8,9].

In the distal portion of the gastrointestinal tract, the microbiota plays a crucial role in two essential metabolic functions related to conjugated and unconjugated BAs that escape reabsorption: the deconjugation of glycine- and taurine-conjugated bile acids; and the conversion of primary to secondary fecal unconjugated bile acids (fUBA) [9]. Several bacterial species deconjugate glycine- and taurine-conjugated BAs through bile salt hydrolase (BSH) activity [10,11]. However, only a few bacterial species have been identified as responsible for converting primary fUBAs (i.e., CA and CDCA) into secondary fUBA, deoxycholic acid (DCA), and lithocholic acid (LCA) [7] through the 7α-dehydroxylation pathway. Among these bacteria, specialized anaerobic species found in the *Peptacetobacter* genus, previously known as *Clostridium hiranonis*, possess the bile acid-inducible (*bai*) operon, which encodes enzymes responsible for the conversion of primary to secondary fUBA [12,13,14,15].

*Peptacetobacter hiranonis*, also known as *Clostridium hiranonis*, is an anaerobic, spore-forming, and Gram-positive bacterium [15,16]. In dogs, *P. hiranonis* is described as a biomarker for gastrointestinal functionality and is closely linked to maintaining balanced gastrointestinal health [17]. *P. hiranonis* occupies a pivotal role in the conversion of primary to secondary fUBA and is described as the main species with this ability within the canine and feline gastrointestinal microbiome [18,19,20,21]. This conversion involves a multi-step biochemical pathway encoded by the *bai* operon. Initially, the primary fUBA enters the cell through a transporter encoded by the *baiG* gene. Subsequently, it is conjugated to coenzyme A (CoA) by a CoA-ligase encoded by the *baiB* gene. The fUBA, bound to the coenzyme A, undergoes oxidation by a dehydrogenase encoded by the *baiA* gene. Lastly, the *baiH* and *baiCD* encode the enzymes responsible for the dehydroxylation of those molecules [22,23].

Dysmetabolism of BAs has been linked to chronic enteropathy (CE) in dogs [24,25], as well as other chronic inflammatory diseases in humans, including obesity, type 2 diabetes, and liver diseases [26,27]. Disruption of BA metabolism has been associated with reduced *P. hiranonis* abundance in several studies [21] and is commonly observed after antibiotic usage and in dogs with CE [19,24,25]. Although the link between *P. hiranonis* and BA metabolism in dogs has been described [19,25], our understanding of the more complex correlation between them is still evolving and remains not completely elucidated. In this study, we aimed to assess the correlation between (1) *P. hiranonis* and *baiCD* gene abundances, (2) *P. hiranonis* abundance and the relative concentration of secondary fUBA, (3) the *baiCD* gene abundance and relative concentration of secondary fUBA in fecal samples, and (4) to determine the likelihood of other unidentified *bai* operon-carrying bacterial species being responsible for BA conversion in dogs.

## 2. Materials and Methods

### 2.1. Animals

In this retrospective study, leftover fecal samples from a previously published study were used [19]. Canine samples were collected from 24 clinically healthy dogs enrolled under the study protocol approved by the Institutional Animal Care and Use Committee at Louisiana State University (Protocol No. 14-027). Health history (no past gastrointestinal abnormalities or use of antibiotics for the 12 past months), physical examination, complete blood count, and serum chemistry were evaluated. The study design was described in detail by Pilla et al. [19], and crucial information and additional analysis are also stated below.

### 2.2. Sampling

As previously described by Pilla et al. [19], 136 fecal samples were collected at different time points over 84 days, aliquoted, and frozen within 4 h of the collection. Samples were kept at −80 °C for further analysis. The control group (group 1) did not receive intervention, and fecal samples were collected on days 0, 7, 21, and 42. Group 2 received a hydrolyzed protein diet and metronidazole administration orally (15 mg/kg every 12 h) for two weeks, between days 42 and 56, and fecal samples were collected on days 0, 7, 21, 42, 49, 56, 70, and 84. Group 3 only received metronidazole orally (15 mg/kg every 12 h), for two weeks, between days 0 and 14, and samples were collected on days 0, 7, 14, 28, and 42. In this study, we re-categorized the assessed time points from all animals into three groups: those in the absence of antibiotic administration, those during antibiotic administration, and those post antibiotic administration. The categorization into these three groups allowed us to assess the changes in the relative concentration of secondary fUBA, *P. hiranonis*, and *baiCD* gene abundances before, during, and after metronidazole-induced intestinal dysbiosis.

### 2.3. DNA Extraction and qPCR Analysis

The DNA from fecal samples was isolated using the MoBio Power Soil Kit (MO BIO Laboratories, Inc., Carlsbad, CA, USA), following the manufacturer’s instructions. The DNA concentration used for the assays was 5 ng/µL, and the qPCR assay was performed using SsoFast EvaGreen^®^ Supermix (Bio-Rad Laboratories, Hercules, CA, USA) as described by AlShawaqfeh et al. [28]. Two sets of primers were used: one set to detect the *P. hiranonis* species (forward: 5′-AGTAAGCTCCTGATACTGTCT-3′; reverse: 5′-AGGGAAAGAGGAGATTAGTCC-3′) [28] and another set to detect *baiCD* gene present in *P. hiranonis* (forward: 5′-GTTGAAGCTGGATTCGATGC-3′; reverse: 5′-ATACCAGCCATACCACCACCGATT-3′). In addition, the abundance of *Clostridium scindens* was assessed using primers (forward: 5′-CTCCGCTGTTCGGTATGGA-3′; reverse: 5′-GCATCGTCATATCCCAGGTCTT-3′) described by Alexander et al. [29].

### 2.4. Specificity of the baiCD Primer and Calibration Curve

The specificity of the *baiCD* primer was assessed via agarose gel electrophoresis. The *baiCD* amplicon was extracted from the gel using QIAquick^®^ Gel Extraction Kit (QIAGEN, Hilden, Germany) and ligated to a pCR^®^ 4-TOPO^®^ vector (Invitrogen^TM^ Life Technologies). Subsequently, the vector transformed the competent DH5α-T1^R^
*Escherichia coli* TOPO^TM^ TA Cloning^TM^ Kit (Invitrogen^TM^ Life Technologies, Carlsbad, USA). The plasmid containing the amplicon was purified using a QIAprep Spin Miniprep Kit (QIAGEN) and confirmed through a conventional PCR assay and by Sanger Sequencing at Eton Bioscience, Inc. (San Diego, CA, USA).

Using a ten-fold dilution of the purified plasmid, the standard curve for the DNA quantification was conducted. The log amount of DNA (number of copies) per 10 ng isolated from total DNA was used to express the qPCR results. The amplicon length, melting peak temperature, efficiency of the qPCR assay, and the coefficient of determination (R^2^) of the calibration curve are summarized in Appendix A. Additionally, besides detecting the *baiCD* amplicon and confirming its size, we assessed the specificity of our primers against different bacterial species including bile acid converter *C. scindens* (wild-type strain) and other intestinal bacteria such as *E. coli* (ATCC^®^ 25922), *Faecalibacterium duncaniae* (DSM 17677), *Akkermansia muciniphila* (ATCC^®^ BAA-835), and *Clostridium difficile* (wild-type strain). The experimental data are summarized in Appendix A.

### 2.5. Fecal Bile Acid Analysis

The concentration of primary fUBA (CA and CDCA) and secondary fUBA, (LCA, DCA, and ursodeoxycholic acid (UDCA)) were measured using the gas chromatography with mass spectrometry method (GC-MS), as previously described by Blake et al. [25]. Data were primarily reported in micrograms per milligram of lyophilized feces (µg/mg of fecal dry matter) and transformed to a percent of the total fUBA measured. Total primary fUBA was defined as the sum of CA and CDCA, and total secondary fUBA as the sum of LCA, DCA, and UDCA. Finally, the total fUBA constitutes the sum of all evaluated fUBA (CA, CDCA, LCA, DCA, and UDCA), and the percentages (%) of primary and secondary fecal unconjugated bile acids were measured based on the total concentration of fUBA in the fecal dry matter (Appendix A).

### 2.6. Statistical Analysis

Shapiro–Wilk’s test was used for the normality assessment of the data. Spearman’s test was employed to assess the correlation between the abundance of *P. hiranonis* and *baiCD* gene, as well as the relative concentration of secondary fUBA in all evaluated samples from groups 1, 2, and 3 combined into one single group (GraphPad Prism 9.4.1). The effects of the hydrolyzed protein diet on the assessed variables (secondary fUBA relative concentration and abundances of both *P. hiranonis* and the *baiCD* gene) were measured by comparing multiple time points to day 0 (baseline). Additionally, the effects of the metronidazole administration on the assessed variables (secondary fUBA relative concentration and abundances of both *P. hiranonis* and *baiCD* gene) were measured by comparing multiple time points (days 49, 56, 70, and 84) to day 42 in group 2, and to day 0 in group 3. The effects of the hydrolyzed protein diet and metronidazole administration were assessed through Friedman’s test adjusted for Dunn’s multiple comparison (GraphPad Prism 9.4.1)**.** A *p*-value of less than 0.05 was considered statistically significant.

## 3. Results

### 3.1. Abundances of P. hiranonis, baiCD Gene, and C. scindens

The abundance of *P. hiranonis* and the *baiCD* gene decreased during and after antibiotic administration. A drastic reduction was observed during the period when the animals were taking metronidazole orally (Table 1). From 133 fecal samples, 113 had leftover extracted DNA available for the quantification of *C. scindens* via qPCR. *C. scindens* were not detected in any of the evaluated samples (0%; 0/113)

### 3.2. Fecal Unconjugated Bile Acids

The relative concentration of secondary fUBA (LCA and DCA) decreased during the period when animals were receiving the antibiotic (Figure 1A), following a pattern observed in the abundances of *P. hiranonis* and *baiCD* gene (Figure 1B,C, respectively).

### 3.3. Correlation Analysis between P. hiranonis, the baiCD Gene, and the Relative Concentration of Secondary Fecal Unconjugated Bile Acids

Spearman’s rank correlation was computed to assess the relationships between *P. hiranonis*, the *baiCD* gene, and the relative concentration of secondary fUBA; all groups were combined for the correlation analysis. Positive correlations were observed between *P. hiranonis* and *baiCD* abundances (ρ = 0.8230, 95% CI (0.7570, 0.8724), *p* < 0.0001), as well as between *P. hiranonis* and the relative concentration of secondary fUBA (ρ = 0.6658, 95% CI (0.5555, 0.7532), *p* < 0.0001). Also, *baiCD* gene abundance showed a positive correlation with the relative concentration of secondary fUBA (ρ = 0.7377, 95% CI (0.6461, 0.8084), *p* < 0.0001) (Figure 2 and Figure 3). Animals displaying high levels for the conversion of primary to secondary fUBA and lacking *P. hiranonis* were not observed (red-colored area in Figure 2B).

### 3.4. Effects of Metronidazole Administration and Hydrolyzed Protein Diet on P. hiranonis, the baiCD Gene and Relative Concentration of Secondary Fecal Unconjugated Bile Acids

The effects of the oral metronidazole and hydrolyzed protein diet on the assessed variables (secondary fUBA relative concentration and abundances of both *P. hiranonis* and *baiCD* gene) were measured by comparing the multiple time points to the baseline time point using Friedman’s test adjusted for Dunn’s multiple comparison. This analysis was performed at multiple time points for dogs within the same group. There were no significant changes observed in group 1 or in group 2 during the dietary trial, except on day 7 when the relative concentration of secondary fUBA was slightly higher compared to day 0 (day 0 median: 91.40%; day 7 median: 97.81%), probably influenced by an outlier (data shown in Figure 3). During metronidazole administration in group 2, the secondary fUBA relative concentration and abundances of both *P. hiranonis* and the *baiCD* gene were significantly reduced during days 49 and 56 compared to the day before metronidazole administration (day 42; *p* < 0.05). Similarly, metronidazole administration significantly reduced secondary fUBA relative concentration, *P. hiranonis*, and *baiCD* on days 7 and 14 when compared to day 0 in group 3 (*p* < 0.05).

## 4. Discussion

The correlation between *P. hiranonis*, *baiCD*, and secondary fUBA demonstrates a strong association between the conversion of BA and the abundance of *P. hiranonis*. Recent studies highlighted the role of *P. hiranonis* as the main species in dogs and cats presenting the *bai* operon, as evidenced by a strong correlation between *P. hiranonis* abundance and the relative concentration of secondary fUBA [30]. This is similar to humans, where *C. scindens* appears to be the major player in BA conversion [31,32].

Unlike many other metabolic pathways, where distinct bacterial species share the same function (for example, the bile salt hydrolase activity), the *bai* operon seems to be unique and scarcely found in the bacteria kingdom. The BSH, also known as *choloylglycine hydrolase*, catalyzes the hydrolysis of amino acid-conjugated BAs into unconjugated BAs. *Lactobacillus*, *Bifidobacterium*, *Enterococcus*, *Clostridium*, and *Bacteroides* are known to express BSH activity, exhibiting variability in catalytic efficiency, substrate preference, and the number of the gene copies owned by each bacterial strain [10]. On the other hand, the *bai* operon has only been described in a few species from the *Clostridiaceae*, *Lachnospiraceae*, *Peptostreptococceae*, and *Ruminococcaceae* families; however, it still exerts a significant effect on both healthy and diseased states in animals, including humans, due to playing an important role in the bile acid metabolism [22,33].

In our study, dogs exhibiting higher levels of secondary fUBA relative concentration and lacking *P. hiranonis* and *baiCD* were not observed (red-colored area in Figure 2B), except for a single sample that had *P. hiranonis* abundance of 4.8 log DNA, just below the lower reference interval of 5.1 log DNA. This finding may suggest that the true threshold for bile acid conversion is likely slightly lower than the published reference interval [34]. However, *P. hiranonis* abundances within the reference interval were found in animals without the conversion of BAs (blue-colored area in Figure 2B). Similar to other metabolic functions, the *baiCD* gene may be detected, but it may be inactivated or display lower activity levels. It is also possible that bile acid conversion was not detected due to the limitations of the GC-MS assay or that secondary bile acid levels were lower due to faster transit times. Indeed, in the original clinical trial, some owners reported the presence of diarrhea in dogs receiving metronidazole, but unfortunately, fecal scores were not recorded for any of the samples.

Regarding the specificity of the *baiCD* primers, they were designed based on *P. hiranonis* reference sequences available on NCBI (GenBank: AF210152.2). No amplification was observed when tested with a wild-type *C. scindens* strain, which is known to carry the *bai* operon, confirming the species-specificity of our primer set. However, while in silico analysis using a BLASTN search was unable to identify nonspecific annealing, nonspecific amplification can occur in vitro experiments.

Although the hypothesis of conversion from other bacterial species was not excluded in our study, the strong correlation suggests that *P. hiranonis* is the main source of the conversion of BAs through the 7α-dehydroxylation pathway in dogs. Both the *baiCD* gene and the *bai* operon appear to be consistently present across different *Clostridium* species. The *bai* operon is defined as an orthologous operon and has been extensively studied in species such as *P. hiranonis*, *C. scindens,* and *C. hylemonae* [16,23,35]. Despite its significant sequence dissimilarities when compared across different species, its core function remains conserved [23]. In our study, we were unable to detect *C. scindens* in any of the samples tested. All samples presenting secondary fUBA within the reference interval were positive for *P. hiranonis* within or slightly below its reference interval (Figure 2B, red area). However, not all samples presenting *P. hiranonis* within the reference interval had secondary fUBA within the reference interval (Figure 2B, blue area). It is still unknown whether every strain of *P. hiranonis* possesses a complete, functional *bai* operon. Therefore, while the role of other bacterial species carrying the bai operon cannot be ruled out, our results show no evidence that they play a significant role in bile acid conversion in dogs.

Reduced secondary fUBA levels have been found in puppies with physiological dysbiosis due to an immature microbiome in the first few months of life, in dogs with CE, and in dogs receiving antibiotics such as tylosin and metronidazole [19,24,30]. In all those studies, *P. hiranonis* abundance was found to be reduced. Furthermore, in humans, an increased abundance of primary fUBA has been reported in irritable bowel syndrome [36,37]. In our study, we chose to include animals undergoing metronidazole administration, a common trigger that induces severe long-lasting dysbiosis. All dogs exhibited a significant reduction in the relative concentration of secondary fUBA over the two weeks of antibiotic administration. As the conversion of primary to secondary fUBA is a metabolic pathway present exclusively in bacteria, particularly in *Clostridium* species, those species sensitive to metronidazole experienced a substantial decline in their abundance and metabolic functions. Consequently, some of these animals did not regain *P. hiranonis* or recover the BA conversion by the last time point of this study due the antibiotic use (Figure 1A).

The hydrolyzed protein diet did not affect secondary fUBA relative concentration or *P. hiranonis* and *baiCD* abundances. Hydrolyzed protein diets are recommended for dogs exhibiting food sensitivities or allergies, and they did not impact the gut microbiome of the dogs included in this study (previously published results [19]). Similarly, other types of hypoallergenic diets have also been shown to not affect the gut microbiome of healthy dogs [38]. Inversely, metronidazole administration had a major impact on microbiome composition in the animals included in this study, which included decreased alpha-diversity, major shifts in beta-diversity, and changes in abundance of taxa in all taxonomic levels (previously published results by Pilla et al. [19]. A significant reduction in fUBA conversion, *P. hiranonis*, and *baiCD* was observed in our study, in agreement with previously published data in which animals were treated with antibiotics [19,24,30].

Among the limitations present in our study, we chose to use healthy dogs with antibiotic-induced dysbiosis due to the consistent and severe changes in the microbiota observed in those animals treated with metronidazole. However, antibiotic-induced dysbiosis may lead to slightly different changes in the intestinal microbiota composition when compared to CE. In dogs, CE is typically subdivided into phenotypes based on the animal’s response to the therapy. Typically, these animals exhibit more pronounced degrees of changes—for example, increased permeability and increased production of pro-inflammatory cytokines—due to the chronic nature of this disease, which is not observed in antibiotic-induced dysbiosis in healthy animals [39]. Furthermore, the sample size can be considered a limitation in this study. While the strong correlation between *P. hiranonis*, *baiCD*, and secondary fUBA was confirmed through statistical methods, it is important to note that multiple time points from the same animals were used to assess the correlation between our variables. Moreover, the in vitro bile acid conversion ability of *P. hiranonis*, previously described in [12,14,23], was not assessed in this study due to the difficulty in isolating *P. hiranonis* from frozen samples after long periods of storage.

Bile acid conversion seems to be a field that requires further development in veterinary medicine. Bile acid conversion in vivo is an intricate and complex metabolic process that can involve several different pathways, such as the 3α-hydroxysteroid dehydrogenase, 7α-hydroxysteroid dehydrogenase, 12α-hydroxysteroid dehydrogenase, and 7 α-dehydroxylation pathways [40]. This study allowed us to have a look into the conversion of bile acids performed by the 7 α-dehydroxylation pathway and the correlation between the conversion of bile acid and *P. hiranonis* abundance. Moreover, the implications of the potential impacts of different forms of bile acids on canine health need further investigation. Since the conversion of secondary bile acids exhibited a strong correlation with the abundance of *P. hiranonis*, *P. hiranonis* seems to be an indicator for bile acid conversion performed by the 7 α-dehydroxylation pathway in vivo. However, more studies are needed to elucidate the predictive value of *P. hiranonis* and *baiCD* abundance in bile acid conversion in vivo for dogs and other species.

Although the *bai* operon has been described in *P. hiranonis* [23], a more comprehensive understanding of its functional role and its relationship with healthy and diseased states in animals is needed. The secondary fUBA production pathway seems to be conserved in dogs, cats, and humans [30,41]; however, understanding its effects on gut homeostasis in different species is necessary. Differences in the conversion of primary into secondary fUBA and other BA-associated compounds have been identified in animals and a better comprehension of its physiological functions and effects on intestinal health needs further research. While this study has provided valuable insights into the role of *P. hiranonis* in the conversion of primary to secondary fUBA and suggests that *P. hiranonis* is the main source of conversion of BAs for this metabolic pathway in dogs, future research could help us to develop a better understanding of the roles of microbiota and BA metabolism in diseases.

## 5. Conclusions

In summary, our study confirms the strong correlation between *P. hiranonis* and *baiCD* abundances, suggesting their ability to predict the conversion of primary to secondary fUBAs in dogs through the 7 α-dehydroxylation pathway. Given the strong correlation between *P. hiranonis* and *baiCD* abundances, the absence of detectable *C. scindens* in our samples, and the absence of samples presenting secondary fUBA within the reference interval but lacking *P. hiranonis*, it is unlikely that other bacterial species play a significant role in BA conversion through the 7 α-dehydroxylation pathway in dogs. Finally, considering the correlation between decreased secondary fUBA, intestinal dysbiosis, and chronic enteropathy in dogs, further studies are needed to evaluate its potential as a target for microbiome- and metabolite-modifying therapeutics.

## Figures and Tables

**Figure 1 animals-14-00216-f001:**
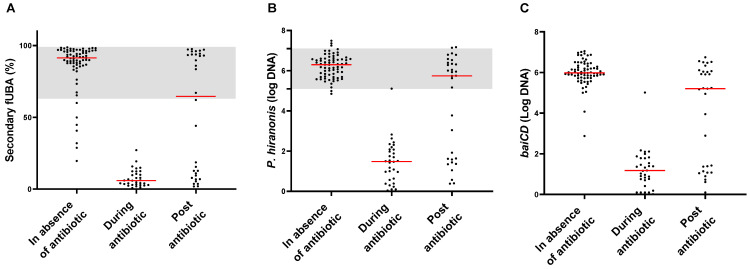
Multiple time points from 24 dogs demonstrating the correspondence of the relative concentration of secondary fUBA (**A**) and the abundances of *P. hiranonis* (**B**) and *baiCD* (**C**) in healthy dogs in the absence of, during, and after antibiotic administration. *P. hiranonis* is the main species responsible for the conversion of primary into secondary fecal unconjugated bile acids in dogs. As shown in these figures, the decrease in *P. hiranonis* abundance is followed by a reduction in *baiCD* abundance and the relative concentration of secondary fUBA. The red lines on the graph represent the median value for each one of the evaluated groups, and the shaded areas indicate the reference intervals for *P. hiranonis* abundance in fecal samples from dogs.

**Figure 2 animals-14-00216-f002:**
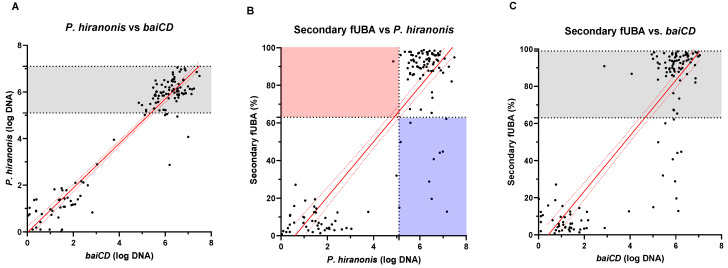
Scatter plots of multiple time points from 24 dogs demonstrating the correlation between the abundances of *P. hiranonis* and *baiCD* and the relative concentration of secondary fUBA. Spearman’s rank correlation analysis revealed strong positive correlations between abundances of *P. hiranonis* and *baiCD* (ρ = 0.8230, 95% CI (0.7570, 0.8724), *p* < 0.0001) (**A**), abundance of *P. hiranonis* and relative concentration of secondary fUBA (ρ = 0.6658, 95% CI (0.5555, 0.7532), *p* < 0.0001) (**B**), and *baiCD* and secondary fUBA (ρ = 0.7377, 95% CI (0.6461, 0.8084), *p* < 0.0001) (**C**) in healthy dogs. Dotted lines represent reference intervals for healthy dogs. Animals displaying high levels for conversion of primary to secondary fUBA and lacking *P. hiranonis* were not observed (red-colored area in (**B**)), except for a single sample that had *P. hiranonis* abundance of 4.8 log DNA, just below the lower reference interval of 5.1 log DNA. The blue-colored area displays animals with a high abundance of *P. hiranonis* but lacking in fUBA conversion.

**Figure 3 animals-14-00216-f003:**
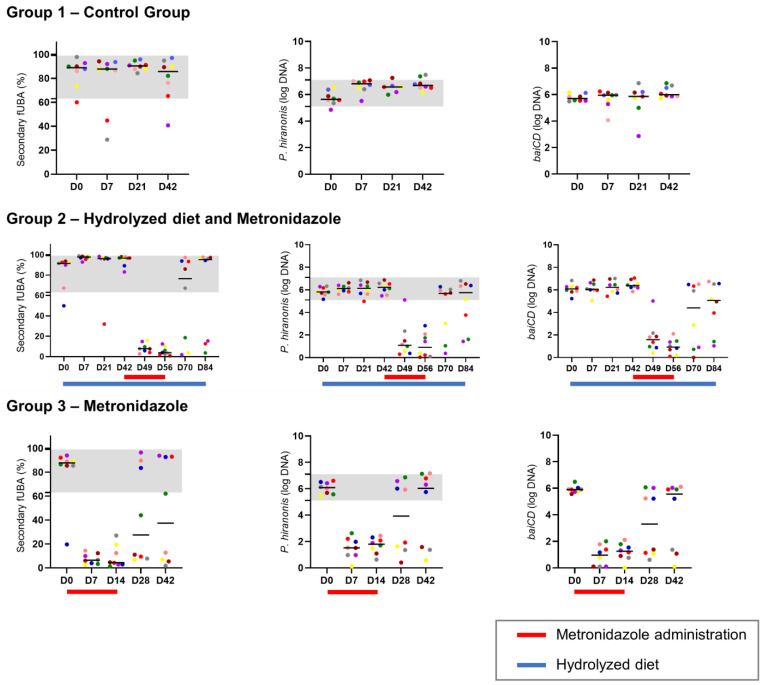
Scatter plots of multiple time points (days; “D”) from 24 dogs randomly assigned to three groups: group 1 (control) did not receive any intervention, group 2 received a hydrolyzed protein diet followed by metronidazole orally, and group 3 received only metronidazole orally. Animals were color-coded individually within their group. The red lines under the graphs represent the period that animals were taking metronidazole orally, while the blue lines represent the period that animals were receiving the hydrolyzed protein diet. The shaded areas indicate the reference intervals for healthy dogs.

**Table 1 animals-14-00216-t001:** Descriptive statistics of *P. hiranonis* and *baiCD* gene abundances expressed in log DNA (median [range]) assessed at multiple time points in dogs in the absence of, during, and after antibiotic administration.

Groups	*P. hiranonis* (log DNA)	*baiCD* Gene (log DNA)
In the absence of antibiotic administration	6.3 [4.8–7.5]	5.9 [2.9–7.0]
During antibiotic administration	1.5 [<0.1–5.1]	1.2 [<0.1–5.0]
Post-antibiotic administration	5.7 [<0.1–7.2]	5.2 [<0.1–6.6]

Note: abundances expressed in log DNA (median [range]).

## Data Availability

All data supporting the findings of this study are available within the paper and its Appendix A.

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
