# Peer review of "Correlation between Peptacetobacter hiranonis, the baiCD Gene, and Secondary Bile Acids in Dogs"

_animals, 2024, doi:10.3390/ani14020216_

Round 1

Reviewer 1 Report

Comments and Suggestions for Authors

Paper review: Correlation between Peptacetobacter hiranonis, baiCD gene, 2 and secondary bile acids in dogs

Line 85: it is suggested to write the number twenty four with digits (24).

Line 89: In the manuscript, the methodology is referenced as being detailed in “Pilla et al”. This suggests that the reader should consult the specified bibliography to fully understand the methods employed. I recommend refining this section for clarity, as the methods are subsequently explained in the manuscript itself.

Line 162-165: The title of Table 1 is appropriately concise up to line 162. Beyond this point, the title becomes excessively lengthy, incorporating details that would be more suitably presented within the text rather than in the title itself. Please consider revising it for brevity and clarity.

Introduction

The introduction is detailed and informative, providing a solid foundation for understanding the study´s context and significance.

Materials and methods

This section is well constructed, with a high level of detail and technical accuracy, wich are critical for the validity and reproducibility of the research. Enhancements in simplifying the language and providing more context in certain areas (like what I said in line 89) could further improve clarity.

Results

The results are presented in a clear and concise manner, with the use of tables and figures to effectively illustrate the data. The findings are directly linked to the study objectives.

Discussion

This section could further explore the broader implications of the study, such as potential impacts on dog health and treatments strategies, to make the findings more relevant to a wider audience, including veterinary practitioners.

Conclusion

Concise and directly related to the study´s objectives. It succinctly summarizes the main findings and their implications.

It could be improved by briefly mentioning the study´s significance or implications for dog health.

Author Response

Answer: Dear reviewer,

Firstly, I would like to thank you for dedicating your time to reviewing our manuscript. I strongly appreciate your input, and I am hopeful that the revisions we have implemented to our manuscript will contribute to its publication acceptance.

In addition, I would like to emphasize that the authors endorse the suggestions you have provided, and, over the past week, we have been working, intending to improve the quality of our manuscript.

Finally, by the reviewer’s suggestions, changes in the methods section (lines 87-165) have been incorporated into our manuscript. The line number was described in simple markup in tracking changes mode.

Thank you once again for your valuable feedback.

Line 85: it is suggested to write the number twenty-four with digits (24).

Answer: Thank you for your suggestion; we have addressed your suggestion in our manuscript (Line 89).

Line 89: In the manuscript, the methodology is referenced as being detailed in “Pilla et al”. This suggests that the reader should consult the specified bibliography to fully understand the methods employed. I recommend refining this section for clarity, as the methods are subsequently explained in the manuscript itself.

Answer: The authors appreciate your attention to this matter, and we edited the text accordingly (lines 93-95). While the detailed methods have been described elsewhere, the authors believe that the reader should be able to understand the current manuscript without having to read our previous study. Therefore, any details crucial for the comprehension of the current manuscript have been kept in the updated version.

Line 162-165: The title of Table 1 is appropriately concise up to line 162. Beyond this point, the title becomes excessively lengthy, incorporating details that would be more suitably presented within the text rather than in the title itself. Please consider revising it for brevity and clarity.

Answer: Thank you for pointing this out; we have removed the unnecessary and excessively lengthy section from the title of Table 1. The changes can be found in lines 174-176.

Introduction: the introduction is detailed and informative, providing a solid foundation for understanding the study´s context and significance.

Answer: We deeply appreciate your review of our paper.

Materials and methods: this section is well constructed, with a high level of detail and technical accuracy, which are critical for the validity and reproducibility of the research. Enhancements in simplifying the language and providing more context in certain areas (like what I said in line 89) could further improve clarity.

Answer: Thank you for pointing out that our methodology section was well-constructed and described in detail. Moreover, the authors agree with the reviewer’s input regarding improving clarity and simplifying the language in the methodology section and changes were incorporated into our manuscript (lines 87-165).

Results: the results are presented in a clear and concise manner, with the use of tables and figures to effectively illustrate the data. The findings are directly linked to the study objectives.

            Answer: Thank you for your comment.

Discussion: This section could further explore the broader implications of the study, such as potential impacts on dog health and treatment strategies, to make the findings more relevant to a wider audience, including veterinary practitioners.

Answer: The authors agree with the reviewer and changes exploring broader implications, potential impacts on dog’s health, and treatment strategies were incorporated into our manuscript (lines 326-350). In addition to these incorporated changes, we would like to mention that potential impacts on dog’s health were also briefly mentioned previously in lines 74-85, 286-299, and 311-325.

Conclusion: Concise and directly related to the study´s objectives. It succinctly summarizes the main findings and their implications. It could be improved by briefly mentioning the study´s significance or implications for dog health.

Answer: Thank you for your input on our manuscript. The authors are in accordance with the reviewer’s suggestions and changes regarding the study’s significance and implication were added to our conclusion section (lines 352-361).

Reviewer 2 Report

Comments and Suggestions for Authors

General comments:  This work describes the results of analysis of concentrations of fecal secondary unconjugated bile acids (fUBA) and the abundance of P. hiranonis and the baiCD operon in the feces of healthy dogs subjected to experimental dysbiosis induction by treatment with metronidazole.  The paper reports a significant positive correlation between the abundance of both P. hiranonis and baiCD and secondary fUBA concentrations in metronidazole treated animals, compared to pre-treatment values. These findings indicate a positive correlation, but do not alone indicate that P. hiranonis is the primary converter of primary to secondary BA in the canine gut, and the last sentence of the Abstract should be reworded. An alternative explanation for the lack of correlation of fUBA and baiCD in some animals, namely that baiCD was not the only gene responsible for BA conversion in all dogs, was not included and should have been mentioned. The limited scope of the paper also limits the potential impact of the findings.

Limitations of the study design and data analysis include lack of clarity on how feeding the hypoallergenic diet impacted fUBA, as the metron alone and diet + metron groups appear to have been combined. Direct evidence of P. hiranonis ability to convert primary to secondary BAs in vitro using pure cultures of P. hiranonis was not provided, but instead inferred.  

Specific comments: 

1.  Figure 1.  Are the animals displaying consistently low fUBA concentrations following cessation of antibiotic therapy the same animals with reduced concentrations of P. hiranonis and baiCD concentrations? 

2.  Were there available data from healthy dogs not subjected to metron treatment, but monitored over the same time frame, to address the issue of how stable fUBA concentrations and P. hiranonis abundances were over time in healthy animals?

3.  What was the impact of feeding the hypoallergenic diet on the 3 parameters measured, compared to metron treatment alone?  The data for the two groups (ie, metron alone, metron + diet) should be plotted individually to allow readers to discern the impact of diet on the 3 parameters evaluated. 

Author Response

Answer: Dear reviewer,

Firstly, I would like to thank you for dedicating your time to review our manuscript. I highly value and appreciate your input, and I am hopeful that the revisions we have implemented to our manuscript will contribute to its publication acceptance.

Additionally, I would like to emphasize that the authors endorse the suggestions you have provided to make our manuscript better. Over the past week, we have been working on your revisions, intending to improve our manuscript's quality.

In summary, we would like to highlight that we have implemented changes in the introduction, methods, results, and conclusion sections as suggested by the reviewer. The line number was described in simple markup in tracking changes mode.

Thank you once again for your valuable feedback.

General comments:  This work describes the results of the analysis of concentrations of fecal secondary unconjugated bile acids (fUBA) and the abundance of P. hiranonis and the baiCD operon in the feces of healthy dogs subjected to experimental dysbiosis induction by treatment with metronidazole. The paper reports a significant positive correlation between the abundance of both P. hiranonis and baiCD and secondary fUBA concentrations in metronidazole-treated animals, compared to pre-treatment values.

Answer: The authors would like to thank you for your thorough review and input on our manuscript. We want to clarify that we have identified a strong, positive, and significant correlation between the abundance of P. hiranonis, the baiCD gene, and secondary fecal unconjugated bile acids in dogs – both before, during, and after treatment with metronidazole combined into a single group. We have incorporated a few changes to our manuscript to ensure this information is clearly stated in our methods and results section (lines 105-110, 152-155, 185-190, and 193-201).

Furthermore, we wish to address the use of the abbreviation “fUBA”. It is important to note that “fUBA”, which stands for “fecal unconjugated bile acids”, was employed for both primary and secondary forms of fecal unconjugated bile acids and not exclusively to describe secondary fecal unconjugated bile acids, as mentioned by the reviewer. To distinguish between primary and secondary forms, we have designated the term 'primary fUBA' when referring to primary forms of fecal unconjugated bile acids and 'secondary fUBA' when discussing the secondary forms. We purposefully chose not to add a second abbreviation for secondary fUBA to prevent any confusion for the reader.

These findings indicate a positive correlation but do not alone indicate that P. hiranonis is the primary converter of primary to secondary BA in the canine gut, and the last sentence of the Abstract should be reworded.

Answer: We appreciate your comment, and we have reworded the concluding sentence in our abstract (lines 30-31) according to your suggestion.

Firstly, we would like to reiterate that animals exhibiting bile acid conversion without detectable levels of P. hiranonis and baiCD were not identified. To make this information easier to visualize, we have added a red-colored area in Figure 2. B (lines 202) and have referred to it throughout the manuscript. This observation and the strong correlation between the assessed variables (P. hiranonis and baiCD abundance and secondary fecal unconjugated bile acids relative concentration) make a strong case that P. hiranonis is an essential species in the bile acid conversion process for dogs.

Secondly, it is noteworthy that in vitro strain capabilities for converting primary into secondary bile acids through the 7 α-dehydroxylation pathway was established in only a limited number of species, including C. scindens (Marion et al., 2019), P. hiranonis (Hirano et al., 1981), and C. hylemonae (Ridlon et al., 2019). The converting capability of C. scindens and P. hiranonis have been demonstrated as highly efficient when compared to other strains, and C. scindens is considered the recognized and acceptable standard species exhibiting the bile acid-inducible (bai) operon (Wise and Cummings, 2023). Therefore, to strengthen our findings, we performed a qPCR to detect C. scindens in the evaluated animals. In line with previous unpublished data from our laboratory, qPCR analysis revealed no detectable C. scindens abundance in any of the evaluated samples. Information regarding C. scindens qPCR was added to our manuscript in lines 119-121.

Third, addressing the reviewer's comments on the role of P. hiranonis as the main bile acid converter in dogs, we would like to point out that bile acid conversion through the 7 α-dehydroxylation pathway requires the presence of the bai operon. From literature and repository sequences, only the species mentioned above as in vitro BA converters have been described as possessing the bai operon, as mentioned in our discussion section (lines 241-251). In addition, there are published data in the literature to support that P. hiranonis is highly abundant in healthy dogs (Sung et al., 2023; Felix et al., 2022; Blake et al., 2019), as well as to support that other species (also possessing the bai operon like C. scindens) are either not present or scarcely present in healthy dogs (Sung et al 2023). In Sung et al (2023), no C. hylemonae was found in any of the animals included in the study (n=296), therefore we do not believe C. hylemonae can have a significant role in BA conversion in dogs.  As mentioned above, we have found no detectable C. scindens, and have no samples in which P. hiranonis was absent but BA conversion was still detectable. Therefore, while we cannot rule out the contribution of other species, we do not have any evidence suggesting a significant role of either of those species.

In summary, while we cannot conclusively assert that P. hiranonis is the exclusive bile acid converter species present in healthy dogs, we can affirm that a strong correlation between the assessed variables indicates/suggests that P. hiranonis is the main converter species found in dogs. Changes were incorporated into our manuscript to clarify this difference.

Blake AB, Guard BC, Honneffer JB, Lidbury JA, Steiner JM, Suchodolski JS. Altered microbiota, fecal lactate, and fecal bile acids in dogs with gastrointestinal disease. PLoS One. 2019 Oct 31;14(10):e0224454.

Felix AP, Souza CM, de Oliveira SG. Biomarkers of gastrointestinal functionality in dogs: A systematic review and meta-analysis. Animal Feed Science and Technology. 2022 Jan 1;283:115183.

Hirano SE, Nakama RY, Tamaki MI, Masuda N, Oda H. Isolation and characterization of thirteen intestinal microorganisms capable of 7 alpha-dehydroxylating bile acids. Applied and Environmental Microbiology. 1981 Mar;41(3):737-45.

Marion S, Studer N, Desharnais L, Menin L, Escrig S, Meibom A, Hapfelmeier S, Bernier-Latmani R. In vitro and in vivo characterization of Clostridium scindens bile acid transformations. Gut Microbes. 2019 Jul 4;10(4):481-503.

Ridlon JM, Devendran S, Alves JM, Doden H, Wolf PG, Pereira GV, Ly L, Volland A, Takei H, Nittono H, Murai T. The ‘in vivo lifestyle’ of bile acid 7α-dehydroxylating bacteria: comparative genomics, metatranscriptomics, and bile acid metabolomics analysis of a defined microbial community in gnotobiotic mice. Gut microbes. 2020 May 3;11(3):381-404.

Sung CH, Pilla R, Chen CC, Ishii PE, Toresson L, Allenspach-Jorn K, Jergens AE, Summers S, Swanson KS, Volk H, Schmidt T. Correlation between Targeted qPCR Assays and Untargeted DNA Shotgun Metagenomic Sequencing for Assessing the Fecal Microbiota in Dogs. Animals. 2023 Aug 11;13(16):2597.

An alternative explanation for the lack of correlation of fUBA and baiCD in some animals, namely that baiCD was not the only gene responsible for BA conversion in all dogs, was not included and should have been mentioned. The limited scope of the paper also limits the potential impact of the findings.

Answer: Thank you for your comment. We would like to mention that the lack of correlation between secondary fecal unconjugated bile acids, P. hiranonis, and baiCD was discussed in lines 252-264. Importantly, it should be noted that dogs displaying conversion of primary to secondary without the presence of P. hiranonis or baiCD were not detected in this study (Figure 2. B; red-colored area).

Additionally, it is known that bile acid conversion can also occur through different pathways beyond 7α-dehydroxylation, such as 3α-hydroxysteroid dehydrogenase, 7α-hydroxysteroid dehydrogenase, 12α-hydroxysteroid dehydrogenase, among other pathways (Lucas et al., 2021). However, the final products of those reactions do not include deoxycholic acid (DCA) and lithocholic acid (LCA), both of which can only be produced through the 7α-dehydroxylation pathway. Only ursodeoxycholic acid (UDCA) among the fUBA measured in this study can be produced without 7α-dehydroxylation, however, the amounts of UDCA in canine feces are very low and cannot explain the BA conversion observed in our study. Unfortunately, the methodology1 we have used to assess the bile acid conversion in our study only allows the detection of products originating from the 7α-dehydroxylation pathway, e.g., cholic acid (CA) and chenodeoxycholic acid (CDCA) converted to DCA and LCA, respectively. In an ongoing separate study, our group is evaluating the in vitro and in vivo bile acid conversion through the different pathways previously cited (3α-hydroxysteroid dehydrogenase, 7α-hydroxysteroid dehydrogenase, 12α-hydroxysteroid dehydrogenase, and 7 α-dehydroxylation) using a novel assay, but that data is beyond the scope of this study.

Moreover, we would like to mention that besides the fact that the applied methodology only allows the assessment of CA, CDCA, DCA, LCA, and UDCA our aim was solely to assess the bile acid conversion generated from the 7α-dehydroxylation pathway. As mentioned before, P. hiranonis is described in the literature as an important player in bile acid conversion using this pathway (Hirano et al., 1981; Ridlon et al., 2020). Finally, we would like to reiterate that we appreciate your comment on this topic and will take these considerations into account for our future studies.

1Gas chromatography with mass spectrometry method, as previously described by Guard et al. 2019: Guard, B.C.; Honneffer, J.B.; Jergens, A.E.; Jonika, M.M.; Toresson, L.; Lawrence, Y.A.; Webb, C.B.; Hill, S.; Lidbury, J.A.; Steiner, J.M.; et al. Longitudinal assessment of microbial dysbiosis, fecal unconjugated bile acid concentrations, and disease activity in dogs with steroid-responsive chronic inflammatory enteropathy. J Vet Intern Med 2019, 33, 1295-1305, doi:10.1111/jvim.15493.

Hirano SE, Nakama RY, Tamaki MI, Masuda N, Oda H. Isolation and characterization of thirteen intestinal microorganisms capable of 7 alpha-dehydroxylating bile acids. Applied and Environmental Microbiology. 1981 Mar;41(3):737-45.

Lucas LN, Barrett K, Kerby RL, Zhang Q, Cattaneo LE, Stevenson D, Rey FE, Amador-Noguez D. Dominant bacterial phyla from the human gut show widespread ability to transform and conjugate bile acids. MSystems. 2021 Aug 31;6(4):10-128.

Ridlon JM, Devendran S, Alves JM, Doden H, Wolf PG, Pereira GV, Ly L, Volland A, Takei H, Nittono H, Murai T. The ‘in vivo lifestyle’ of bile acid 7α-dehydroxylating bacteria: comparative genomics, metatranscriptomic, and bile acid metabolomics analysis of a defined microbial community in gnotobiotic mice. Gut microbes. 2020 May 3;11(3):381-404.

Limitations of the study design and data analysis include a lack of clarity on how feeding the hypoallergenic diet impacted fUBA, as the metronidazole alone and diet + metronidazole groups appear to have been combined.

Answer: The authors would like to thank you for your comment. As mentioned in the objectives and methodology section, the focus of this study was describing the correlation between fecal unconjugated bile acid conversion, P. hiranonis, and the baiCD gene. While we have included the basic information on study design for the sake of the reader, additional information regarding our study population, including an extensive evaluation of the effects of metronidazole treatment and hypoallergenic diet effects, can be found in this article: Effects of metronidazole on the fecal microbiome and metabolome in healthy dogs - Pilla - 2020 - Journal of Veterinary Internal Medicine - Wiley Online Library).

As reported by Pilla et al. (2020), the hypoallergenic diet did not affect alpha diversity – both richness and evenness – when evaluating the animals across various time points using the analysis of 16S rRNA genes. In addition, changes in beta diversity – diversity between samples – were not statistically significant. When using qPCR to assess the abundance of specific bacterial species (Blautia, Fusobacterium, P. hiranonis, Streptococcus, Escherichia coli, Faecalibacterium, and Turicibacter), only Streptococcus and Escherichia coli were found to increase on day 42 of the hypoallergenic diet in Group 2.

However, since we would like the manuscript to stand alone without forcing the reader to look into the references for clarity, changes have been incorporated into our manuscript to ensure that we have briefly covered the potential impact of the hypoallergenic diet in the fecal unconjugated bile acids relative concentrations (lines 218-233 and Figure 3 in lines 211).

Direct evidence of P. hiranonis's ability to convert primary to secondary BAs in vitro using pure cultures of P. hiranonis was not provided but instead inferred.  

Answer: The authors appreciate your attention to this matter. The conversion of bile acids by P. hiranonis was first described in 1981 by Hirano et al. and by Doerner et al. in 1997, and posteriorly confirmed by Ridlon in 2020. In addition, sequenced genomes of P. hiranonis available in public repositories carry all the genes from the bai operon necessary to perform the conversion through the 7α-dehydroxylation pathway. This information is included in the manuscript in lines 265-276.

  1. hiranonis was not cultured from the animals evaluated in this study, and neither was the ability of those strains to convert bile acids assessed. That is because due to the retrospective nature of the study, the leftover samples had unfortunately been frozen for a long time, which in our experience is detrimental to P. hiranonis viability. We recognize this as a limitation of our study (lines 323-325). Our research group is currently quantifying the bile acid conversion capabilities of P. hiranonis in a set of isolated canine strains in an effort to identify variations in the efficiency of conversion, and our strains, like those of the literature mentioned above, are also capable of converting BA through the 7α-dehydroxylation pathway.

Finally, this study is a retrospective study based on samples collected by Pilla et al. (2020). P. hiranonis is a strictly anaerobic species, extremely sensitive to oxygen exposure. Culturing P. hiranonis is challenging when performed in fresh samples as described by Vinithakumari et al. (2021) and the culture of P. hiranonis can be extremely challenging in frozen archived feces.

Doerner KC, Takamine F, LaVoie CP, Mallonee DH, Hylemon PB. Assessment of fecal bacteria with bile acid 7 alpha-dehydroxylating activity for the presence of bai-like genes. Applied and environmental microbiology. 1997 Mar;63(3):1185-8.

Hirano SE, Nakama RY, Tamaki MI, Masuda N, Oda H. Isolation and characterization of thirteen intestinal microorganisms capable of 7 alpha-dehydroxylating bile acids. Applied and Environmental Microbiology. 1981 Mar;41(3):737-45.

Ridlon JM, Devendran S, Alves JM, Doden H, Wolf PG, Pereira GV, Ly L, Volland A, Takei H, Nittono H, Murai T. The ‘in vivo lifestyle’ of bile acid 7α-dehydroxylating bacteria: comparative genomics, metatranscriptomic, and bile acid metabolomics analysis of a defined microbial community in gnotobiotic mice. Gut microbes. 2020 May 3;11(3):381-404.

Vinithakumari AA, Hernandez BG, Ghimire S, Adams S, Stokes C, Jepsen I, Brezina C, Sahin O, Li G, Tangudu C, Andreasen C. A model screening pipeline for bile acid converting anti-Clostridioides difficile bacteria reveals unique biotherapeutic potential of Peptacetobacter hiranonis. BioRxiv. 2021 Sep 30:2021-09.

Specific comments: 

  1. Figure 1.  Are the animals displaying consistently low fUBA concentrations following cessation of antibiotic therapy the same animals with reduced concentrations of P. hiranonisand baiCD concentrations? 

Answer: Thank you for your attention to Figure 1. The animals displaying low secondary fecal unconjugated bile acid relative concentrations are the same presenting reduced abundances of P. hiranonis and baiCD gene. We have added a figure with P. hiranonis, baiCD, and secondary fecal unconjugated bile acid relative concentration values color-coded by animals to make this point more evident (Figure 3 in line 209).  

  1. Were there available data from healthy dogs not subjected to metronidazole treatment, but monitored over the same time frame, to address the issue of how stable fUBA concentrations and P. hiranonisabundances were over time in healthy animals?

Answer: The authors appreciate your attention to this matter. We have added a figure (Figure 3 in line 211) containing information regarding the secondary fecal unconjugated bile acid relative concentration values, P. hiranonis, and baiCD abundances over time in healthy dogs in all 3 groups, including group 1 which was the control group (no intervention).

  1. What was the impact of feeding the hypoallergenic diet on the 3 parameters measured, compared to metronidazole treatment alone?  The data for the two groups (i.e., metronidazole alone, metronidazole + diet) should be plotted individually to allow readers to discern the impact of diet on the 3 parameters evaluated. 

Answer: Thank you for your question. The authors have incorporated changes to this manuscript to cover the impact of feeding a hypoallergenic diet and the impact of metronidazole treatment in the three parameters measured in this study (lines 211-233 and Figure 3).  As mentioned in the past two answers, Figure 3 was added to elucidate and clarify questions regarding the three assessed variables: secondary fecal unconjugated bile acids, P. hiranonis, and baiCD.

Reviewer 3 Report

Comments and Suggestions for Authors

The aim of the study by Correa Lopes et al. was to study the correlation between Peptacetobacter hiranonis, the baiCD gene and secondary bile acids in dogs. For this, a metronidazole treatment was used to induce dysbiosis in dogs. The paper is interesting and generally well written, but the findings are very preliminary. The authors found a strong correlation between Peptacetobacter hiranonis, baiCD, and secondary bile acids, using statistical methods. However, changes in microbiota composition and bsh gene abundance (among others) were not assessed in metronidazole-treated animals. Since the BSH reaction is the gateway for bile acid metabolism it is likely that a reduction in BSH microorganisms also contributes to a deacrease in fUBA. Furthermore, the decrease in Peptacetobacter hiranonis is quite expected due to ATB treatment, but as other bacterial populations were not measured, conclusions are highly speculative.

Moreover, the references are very old. There is a lot of updated literature on bile acid metabolism.

Author Response

Comments and Suggestions for Authors

Answer: Dear reviewer,

Firstly, I would like to thank you for spending your time reviewing our manuscript. I strongly appreciate your input and I hope the changes we have incorporated to our manuscript can make it a published article.

Additionally, I would like to emphasize that the authors endorse the suggestions you have provided to make our manuscript better. Over the past week, we have been working on your revisions, intending to improve our manuscript's quality.

By the reviewer’s suggestions, changes in background information, reference list, and conclusions have been incorporated into our manuscript. The line number was described in simple markup in tracking changes mode.

Thank you once again for your valuable feedback.

The aim of the study by Correa Lopes et al. was to study the correlation between Peptacetobacter hiranonis, the baiCD gene, and secondary bile acids in dogs. For this, a metronidazole treatment was used to induce dysbiosis in dogs. The paper is interesting and generally well-written, but the findings are very preliminary. The authors found a strong correlation between Peptacetobacter hiranonisbaiCD, and secondary bile acids, using statistical methods. However, changes in microbiota composition and bsh gene abundance (among others) were not assessed in metronidazole-treated animals. Since the BSH reaction is the gateway for bile acid metabolism it is likely that a reduction in BSH microorganisms also contributes to a decrease in fUBA.

Answer: Thank you very much for your revision. The authors agree that the bile salt hydrolase (bsh) activity is a gateway component in bile acid metabolism, facilitating the conversion of bile acids. It is important to mention that the bsh activity, encoded by the choloylglycine hydrolase gene, is widely distributed in the gut microbiome, and can be also found in P. hiranonis strains as highlighted by Vinithakumari et al. (2021). The choloylglycine hydrolase gene sequences can be easily accessed in published reference genomes for Peptacetobacter hiranonis.

While the significance of the bsh enzyme in bile acid metabolism is emphasized by the reviewer, assessing bsh activity can be challenging, as demonstrated by Jones et al. (2008).  The bsh gene. being a homologous and strain-dependent gene, exhibits a high level of redundancy in the gut microbiome, noting the substantial number of bacterial species harboring this gene. Therefore, since P. hiranonis is known to carry the BSH gene, lack of bsh activity is unlikely to be a cause of the reduced conversion of bile acids in this study.

Finally, we are not measuring conjugated fecal bile acids in this study. The assessment of deconjugation of conjugation bile acids requires a methodology that can detect conjugated bile acids, for example, taurocholic acid or taurochenodeoxycholic acid, among others. Unfortunately, the methodology employed in this study only allows the detection of fecal unconjugated bile acids (fUBA): cholic acid, chenodeoxycholic acid, deoxycholic acid, ursodeoxycholic acid, and lithocholic acid). Therefore, while variations in bsh gene copies and/or activity could influence the pool of fUBA we are able to measure in our assay, it should not impact the ratios calculated as we only considered fUBA on those. We would like to reiterate that our objectives cover only the conversion of bile acids, without mentioning the assessment of the deconjugation step in this pathway.

Lastly, we would like to highlight that a full microbiome analysis was included in the original study (Pilla et al., 2020). That information was mentioned in the manuscript on lines 88-89.

Jones, B.V.; Begley, M.; Hill, C.; Gahan, C.G.M.; Marchesi, J.R. Functional and comparative metagenomic analysis of bile salt hydrolase activity in the human gut microbiome. P Natl Acad Sci USA 2008, 105, 13580-13585, doi:10.1073/pnas.0804437105.

Vinithakumari AA, Hernandez BG, Ghimire S, Adams S, Stokes C, Jepsen I, Brezina C, Sahin O, Li G, Tangudu C, Andreasen C. A model screening pipeline for bile acid converting anti-Clostridioides difficile bacteria reveals unique biotherapeutic potential of Peptacetobacter hiranonis. BioRxiv. 2021 Sep 30:2021-09.

Furthermore, the decrease in Peptacetobacter hiranonis is quite expected due to ATB treatment, but as other bacterial populations were not measured, conclusions are highly speculative.

Answer: Thank you very much for your attention to this matter. The effects of metronidazole treatment on other bacterial populations were covered by the article we have cited in the methodology section in lines 88-89 (Pilla et al., 2020). As described by Pilla et al. (2020), metronidazole treatment changed significantly bacterial composition in groups 2 and 3 with a decrease in bacterial richness and a reduction in Fusobacteria when analyzing the microbiome using the 16S rRNA sequencing. When assessing the bacterial abundances in fecal samples using qPCR, Turicibacter, Fusobacterium, and P. hiranonis were reduced, while Streptococcus and Escherichia coli were increased after metronidazole treatment.

The authors agree that antibiotic treatment can reduce bacterial abundance, including P. hiranonis abundance. In fact, the reason why we chose to assess the correlation in this dataset was precisely so we could cover the whole spectrum of abundances of P. hiranonis, and of concentrations of secondary fUBA. It was not the objective of this study to evaluate the impact of metronidazole in P. hiranonis as that information has already been described in the same dataset: Pilla et al., 2020 (link: Effects of metronidazole on the fecal microbiome and metabolome in healthy dogs - Pilla - 2020 - Journal of Veterinary Internal Medicine - Wiley Online Library).

Finally, we would like to highlight that the objectives of this study were to assess (1) the correlation between P. hiranonis and baiCD gene abundances, (2) P. hiranonis abundance and relative concentration of secondary fecal unconjugated bile acid, (3) the baiCD gene abundance and relative concentration of secondary fecal unconjugated bile acid in fecal samples, and (4) to determine the likelihood of other unidentified bai operon-carrying bacterial species being responsible for bile acid conversion in dogs. The discussion about the effects of metronidazole in different bacterial species is beyond the scope of our manuscript.

Moreover, the references are very old. There is a lot of updated literature on bile acid metabolism.

Answer: Thank you for bringing attention to our reference list. Bile acid metabolism is an aged topic, and we believe it is a best practice to include the original reference for the information being described. However, we have included additional more recent literature on bile acid metabolism in our manuscript, and the authors are open to suggestions from the reviewers regarding our referenced articles.

Round 2

Reviewer 3 Report

Comments and Suggestions for Authors

Accept in present form